# Three Methods Assessing the Association of the Endophytic Entomopathogenic Fungus *Metarhizium robertsii* with Non-Grafted Grapevine *Vitis vinifera*

**DOI:** 10.3390/microorganisms10122437

**Published:** 2022-12-09

**Authors:** Mathilde Ponchon, Annette Reineke, Marie Massot, Michael J. Bidochka, Denis Thiéry, Daciana Papura

**Affiliations:** 1Department of Crop Protection, Hochschule Geisenheim University, 65366 Geisenheim, Germany; 2INRAE, Bordeaux Sciences Agro, ISVV, UMR SAVE, 33140 Villenave d’Ornon, France; 3INRAE, Univ. Bordeaux, UMR BIOGECO, 33610 Cestas, France; 4Department of Biological Sciences, Brock University, St. Catharines, ON L2S 3A1, Canada

**Keywords:** endophytes, rhizosphere, ddPCR, fungal entomopathogens

## Abstract

Characterizing the association of endophytic insect pathogenic fungi (EIPF) with plants is an important step in order to understand their ecology before using them in biological control programs. Since several methods are available, it is challenging to identify the most appropriate for such investigations. Here, we used two strains of *Metarhizium robertsii*: EF3.5(2) native to the French vineyard environment and ARSEF-2575-GFP a laboratory strain expressing a green fluorescent protein, to compare their potential of association with non-grafted grapevine *Vitis vinifera*. Three methods were used to evaluate the kinetics of rhizosphere and grapevine endospheric colonization: (i) Droplet Digital (ddPCR), a sensitive molecular method of *M. robertsii* DNA quantification in different plant parts, (ii) culture-based method to detect the live fungal propagules from plant tissues that grew on the medium, (iii) confocal imaging to observe roots segments. Both strains showed evidence of establishment in the rhizosphere of grapevines according to the culture-based and ddPCR methods, with a significantly higher establishment of strain EF3.5(2) (40% positive plants and quantified median of exp(4.61) c/μL) compared to strain ARSEF-2575-GFP (13% positive plants and quantified median of exp(2.25) c/μL) at 96–98 days post-inoculation. A low incidence of association of both strains in the grapevine root endosphere was found with no significant differences between strains and evaluation methods (15% positive plants inoculated with strain EF3.5(2) and 5% with strain ARSEF-2575-GFP according to culture-based method). ddPCR should be used more extensively to investigate the association between plants and EIPF but always accompanied with at least one method such as culture-based method or confocal microscopy.

## 1. Introduction

Endophytic insect pathogenic fungi (EIPF) support plant health in multiple ways and are known as plant biological control agents against pests [1], plant growth stimulators [2] and plant vaccines [3]. Besides their pathogenic nature, most of these attributes stem from their rhizosphere competence and endophytic potential with diverse plant species even if the underlying mechanisms are not well elucidated [4]. Indeed, EIPF are able to establish inside root tissues without causing significant symptoms of infection in plants [5,6] while simultaneously surrounding the root surface [7]. *Metarhizium robertsii* (Metchnikoff) Sorokin (1883) is an important species used as a commercial microbial biological control agent because of its direct parasitism toward insect pests [8], its potential to colonize the plant rhizosphere [9,10] as well as its ability to establish as an endophyte [11,12,13]. Its root colonization potential is an important feature in order to optimize its capacity as a biological control agent and as a stimulator of plant health.

Efficient and practicable modes of inoculation are a prerequisite to successfully use EIPF as endophytes in crop plants. Previous studies have evaluated and compared techniques for EIPF endophytic inoculation in various plants, such as stem injection [14], leaf spraying [15], root drenching [16], root dipping [17,18] and seed dressing [19,20]. These techniques allowed the characterization of EIPF inoculation success, the respective plant organs colonized, and the extent of colonization [2,6]. However, the dynamics of EIPF establishment inside inoculated plants needs further investigation as very little is known about fungal behavior inside the respective plant and its propensity to colonize within the plant. Moreover, important research gaps still exist regarding the process of endophytic colonization by EIPF, including insights in the endophytic path of entry into the plant tissue, the precise cartography of colonized areas of plants organs as well as distinct tissue colonization. 

A panel of methods is available to evaluate endophytic colonization of EIPF. The most common is the culture-based method, which consists of plating excised plant pieces or a homogenate of surface-sterilized plant tissues from inoculated plants on a growing medium containing antibiotics and to morphologically characterize the microorganism growing on the plate after incubation [21]. This method has the advantage of being affordable, easy to handle, and rapid if the fungus grows relatively quickly on respective plates [22]. However, quantification of fungal propagules via culture-based method has a high degree of imprecision [23]. Moreover, the magnitude of endophytic colonization of plants can be overestimated by the culture-based method. Indeed, it allows quantification of the rate of colonization of the sampled plant area, and, by sampling several plant areas, to characterize the total percentage of colonized plant surfaces [16]. However, these measurements are only partially reliable as the mean percentage of plant area colonization can be overestimated by a single plant or a single organ being totally colonized. 

Microscopic techniques, including confocal imaging, are highly appropriate for such studies as they constitute an empirical method available to characterize the endophytic colonization of the plant and the rhizosphere [24]. They provide visual proof of colonization with the advantage of giving clear information of the proportion of samples and tissues being colonized by respective fungal structures [6]. However, microscopic methods are labor intensive and require exhaustive screening of respective plant organs and tissues [7]. As the extent of endophytic colonization also depends on the respective fungal strain, the host plant [12] and the environment from which the strain was sampled [25], applying a more reliable method for calculating the percentage of EIPF colonization of plants would improve identification of suitable EIPF strains.

Droplet digital PCR (ddPCR^TM^) is a molecular PCR-based method allowing absolute quantification of targeted DNA even in very minimal amounts with better resolution than quantitative PCR (qPCR) [26]. Its specificity and sensitivity are key features of this method, which in turn is important for targeting endophytes as they are present in small quantities in plant organs and other microbial taxa could, as well, be amplified in standard qPCR in case of poor sensitivity [21,24,27]. As an example, ddPCR was proven efficient and more precise than qPCR to quantify inoculum of *Ilyonectira liriodentri* in samples of soil, rhizosphere and grapevine rhizoendosphere [28]. However, ddPCR quantifies DNA from viable and non-viable cells and thus cannot assess the viability of the quantified inoculum. As a consequence, this method should be used together with a method that allows an estimation of the viability of the fungal propagules quantified such as the culture-based method.

Proof of endophytic colonization of *M. robertsii* was found in diverse plants from various botanical families such as tomato *Solanum lycopersicum* L., soy *Glycine max* L., wheat *Triticum aestivum* L., the broad bean *Vicia faba* L., cabbage *Brassica oleracea* L., and the French bean *Phaseolus vulgaris* L. [24,29,30,31,32,33]. However, it was never investigated in domesticated grapevine *Vitis vinifera*, which is a perennial dicot plant grown grafted in most commercial vineyards worldwide [34]. Grafting stands as the most efficient biological control solution against one of the main soil-borne pests, the grapevine phylloxera (*Daktulosphaira vitifoliae*). However, some global regions still grow grapevines non-grafted, such as Argentina and Australia [35,36] necessitating costly quarantine measures to protect the plants. For this reason, protecting own-rooted grapevine through endophytic association with *M. robertsii* would represent a major advantage for sustainable crop protection.

Grapevine above-ground endophytic fungal communities are of critical importance, especially because of their putative role in preventing fungal diseases, however their diversity and distribution are poorly investigated [37,38]. A close relative of *M. robertsii*, *M. pinghaense* was once assessed as a natural grapevine endophyte of *V. vinifera* cv. Cabarnate Gernischet from a pooled homogenate of all above-ground tissue samples, with speculation that this species could be native to the soil and had colonized grapevine tissues as an endophyte through time [38]. Thus, *M. robertsii* could have an endophytic potential of establishment within grapevine as this species was also found native to vineyard soil all around the world [39,40,41,42,43,44,45]. Some studies investigated the potential of artificially inoculated endophytes in grapevine. The EIPF *Beauveria bassiana* has been shown to colonize grapevine as an endophyte [46], with antagonistic activity against downy mildew *Plasmopara viticola* [47,48], grapevine mealybugs *Planococcus ficus* and leafhopper *Empoasca vitis* [48,49].

The aim of this study was to demonstrate the rhizospheric and endophytic potential of *M. robertsii* with non-grafted grapevine *V. vinifera* after artificial inoculation. Accordingly, we screened the presence of endophytic *M. robertsii* in different parts of the grapevine (root, stem and leaf pieces) using in vitro plants. For this purpose, we used three methods: (i) ddPCR for absolute quantification of minimal amounts of *M. robertsii* DNA inside grapevine tissues, (ii) culture-based method to detect live propagules of *M. robertsii* extracted from plant tissues, (iii) confocal imaging to visually track endophytic *M. robertsii* inside the colonized plant. In this study ddPCR was used for the first time to characterize the association between EIPF and plant. Another aim of this study was to compare the kinetics of grapevine colonization by a *M. robertsii* strain native to a French vineyard environment (EF3.5(2)) and a *M. robertsii* transformant strain (ARSEF-2575-GFP) expressing green fluorescence protein (GFP), which is a laboratory strain originally non-native to the vineyard environment [50]. Accordingly, the goal was to assess whether a strain collected in the same environment as grapevine that we wish to protect, has a better and more durable colonization potential. We hypothesized that (i) *M. robertsii* colonization of grapevine is limited to the roots and occurs more extensively in the rhizosphere compared to the root endosphere, (ii) the strain EF3.5(2) native to the vineyard has higher potential to persistently associate with grapevine rhizosphere and root endosphere compared to the ARSEF-2575-GFP strain.

## 2. Materials and Methods

### 2.1. Fungal Cultures

Two *M. robertsii* strains were used in this study. *M. robertsii* strain EF3.5(2) was collected in the INRAE experimental vineyard soil in spring 2015 (Villenave d’Ornon, South West of France, 44°47′30.4″ N 0°34′36.9″ W) via insect bait technique [51], and was kept as a laboratory culture since then. A previous study has shown its potential as a biocontrol agent against the Asian hornet *Vespa velutina* [39]. The fungal identification was confirmed by morphological analysis and by genetic sequencing of the translation elongation factor 1-a [52]. A transformant of *M. robertsii* (ARSEF-2575-GFP) expressing green fluorescence protein (GFP), originally collected from a coleopteran host insect (*Curculio caryae* [Coleoptera: Curculionidae]) in the United States [53] and maintained after transformation in the ARS Collection of Entomopathogenic Fungal Cultures (ARSEF) (US Plant, Soil and Nutrition Laboratory, Ithaca, NY, USA) [54] was also used for the experiment (courtesy of Prof. Dr. Michael Bidochka and Dr. Shasha Hu). Stock cultures were grown on oat agar chloramphenicol media (40 g organic oat flour, (Moulin Des Moines, Krautwiller, France)), 20 g agar (SIGMA Aldrich, St. Louis, MO, USA), 50 mg chloramphenicol (SIGMA Aldrich), and 1 L of water. Conidia were dislodged from the surface into a sterile suspension solution (1/8 Ringer + 0.02% Tween 80^®^ (SIGMA Aldrich). The conidial suspension was adjusted to 1 × 10^7^ conidia·mL^−1^ using a hemocytometer.

### 2.2. Plant Material

Grapevines *Vitis vinifera* cultivar Cabernet Sauvignon were micro propagated in vitro and acclimated in humid plastic containers for 44 days, as described in [55]. Grapevines harboring an average of 8 newly formed leaves were planted in pots (volume 0.3 L) filled with vineyard soil taken from the same vineyard where *M. robertsii* strain EF3.5(2) was originally isolated, however in a different vineyard plot. The soil was previously sterilized by autoclaving twice. Plants were grown and watered every third day and maintained for 60 days in a growth chamber with a 16:8 photoperiod, 23 °C and 60% relative humidity in the facilities of the INRAE UMR 1065—SAVE.

### 2.3. Fungal Inoculation of Grapevines 

Forty-four potted grapevine plants were inoculated with either *M. robertsii* strain ARSEF-2575-GFP or EF3.5(2) by watering the roots with 50 mL of a 1 × 10^7^ conidia·mL^−1^ suspension split into two doses of 20 mL on the first day and 30 mL on the 6th day. The concentration was judged optimal to successfully establish several strains of EIPF including *Metarhizium* spp. as endophyte with root drenching treatment according to [56]. Twenty-eight control plants were treated with the same volume of a sterile suspension solution.

### 2.4. Rhizospheric Detection and Endophytic Association Assessments of Roots, Leaves and Stems Using Culture-Based Method 

After 14-, 35-, 63-, 96–98- days post-inoculation (dpi), grapevines were uprooted for analysis. For rhizosphere association assessment, grapevines were gently removed from their pots and roots were shaken with forceps to remove adhering soil particles. Roots were not disinfected and soil particles were adherent to these roots, hence we considered these root samples representative of the grapevine rhizosphere. From each plant, an average of 1 g of randomly picked root pieces was cut and placed in a tube (height × width: 60 mm × 27 mm and 20 mL volume, ZINSSER POLYVIALS^®^) with 4 mL of sterile distilled water and 0.02% Tween 80^®^ and mixed with 2 inox balls of 8 mm diameter using the disrupter TissueLyser II (Qiagen, Hilden, Germany). A 100 μL of root homogenate as well as a 10-fold diluted homogenate was spread in duplicates onto a chloramphenicol, thiabendazole, cycloheximide (CTC) medium according to a modified recipe from [57] (39 g·L^−1^ potato dextrose agar, 0.1 g·L^−1^ chloramphenicol diluted in 96% ethanol, 0.002 g·L^−1^ thiabendazole and 0.15 g·L^−1^ cycloheximide each diluted in sterile water (SIGMA Aldrich) and filled up to 1 L with sterile water). The plates were incubated at 25 °C for 14 days. *M. robertsii* colonies were visually identified according to morphological features according to criteria described by [54]. The remaining root homogenate was immediately lyophilized for extraction of DNA. 

For endophytic analysis, 1 g of root pieces was sampled in the same way as described above, as well as the third and the terminal leaf and a 3 cm part of the upper stem from each grapevine plant. All root samples were surface-disinfected by dipping them twice in two different solutions of 0.5% NaOCl and 0.02% Tween 80^®^, followed by 2 min in 70% ethanol and rinsing thrice in sterile water. Leaves and stems were disinfected in the same manner, except that they were dipped only once in 0.5% NaOCl and 0.02% Tween 80^®^ for 2 min. Samples were then processed as described above, except for stem pieces which were entirely lyophilized after sampling.

### 2.5. Quantification of Rhizospheric and Endophytic Association Using Droplet Digital PCR 

DNA from inoculated grapevine samples was extracted with the DNeasy PowerSoil Pro kit (Qiagen, Hilden, Germany) following the manufacturer’s directions. To carry out the molecular absolute quantification, the device QX200 DROPLET DIGITAL PCR (ddPCR^TM^) System (Bio-Rad, Hercules, CA, USA) installed at the Genome Transcriptome Platform of Bordeaux (BIOGECO, Bordeaux, France) was used. The primers used for the PCR reactions were Ma 1763 (5′-CCAACTCCCAACCCCTGTGAAT-3′) and Ma 2097 (5′-AAAACCAGCCTCGCCGAT-3′) designed by [58] positioned in the regions of ITS 1 and ITS 2 of the nuclear ribosomal RNA gene cluster, respectively, which were shown to be specific to *Metarhizium* clade 1 which includes *M. robertsii*. The 22 μL PCR reaction mix per sample was composed of 2 μL DNA, 11 μL of QX200^TM^ ddPCR^TM^ EvaGreen Supermix (containing a dsDNA-binding dye) (Bio-Rad, USA), 2.2 μL of each primer at 150 nM, plus 4.6 μL of pure water. Samples were run as single replicates with a volume of 20 μL of mix per sample. A negative control with 2 μL of ultrapure water, as well as one positive control with fungal DNA of *M. robertsii* strain EF3.5(2) extracted from a pure culture was included on every ddPCR plate. Each 20 μL sampling mix was divided into droplets with the QX200 Droplet Generator (Bio-Rad) and then transferred to a 96-well PCR plate. The thermocycling program was set as [95 °C × 5 min; 40 cycles of (95 °C × 30 s, 61 °C × 1 min), 4 °C × 5 min and 90 °C × 5 min] in the Bio-Rad C1000 (Bio-Rad). The device QX200 droplets reader screened each droplet solely for fluorescent signal. The absolute number of copies of targeted fungal DNA sequence per μL of the sample was calculated with a Poisson model, processing the number of positive droplets out of 20,000 droplets (QuantaSoft^TM^ version 1.7, Bio-Rad software). The threshold defining the detection of the positive droplets was adjusted manually at the value of 5000 of fluorescence amplitude. Finally, the dilution factor of the DNA extract in the reaction mix (2 μL in 22 μL) was used to calculate the ultimate absolute concentration of each sample. When analyzing the data, the final values of quantification that were inferior to 1 copies/μL were considered as null for the analysis.

### 2.6. Observations of Rhizospheric and Endophytic Association Using Confocal Microscopy

Confocal imaging was completed at the Bordeaux Imaging Center (BIC) (Bordeaux University, Bordeaux, France) on a Zeiss LSM 880 confocal laser scanning microscope equipped with fast Airy Scan using Zeiss C PL APO × 63 oil-immersion objective. For GFP, excitation was achieved with a 488 nm laser power and fluorescence emission collected at 505–550 nm. Grapevines used for microscopic observation were produced as described above (see Section 2.2) and 12 grapevines in vitro were inoculated by drenching with 50 mL of fungal suspension of 1 × 10^7^ conidia·mL^−1^ of *M. robertsii* strain ARSEF-2575-GFP. After 14-, 31-, 63- and 98- dpi, respectively, 3 plants were uprooted and analyzed through laser scanning confocal microscopy.

### 2.7. Statistical Analysis

All analysis were carried out in the R software updated version RStudio 2022.07.2 + 576 (2022 RStudio©, PBC. All Rights Reserved). The non-parametric Kruskal–Wallis test was used to determine significant differences (*p* < 0.05) between the two tested strains and the control treatment for the number of DNA copies/μL quantified in the rhizosphere and root endosphere via ddPCR. Finally, the percentage of detection of *M. robertsii* as established in the rhizosphere and root-endosphere was analyzed separately for each strain by comparing results from both ddPCR and culture-based method with the Chi^2^ test.

## 3. Results

### 3.1. Quantification of Association of Two M. robertsii Strains with Grapevines via ddPCR

#### 3.1.1. Quantification of Rhizospheric Potential of Two *M. robertsii* Strains

For the rhizosphere, ddPCR detected *Metarhizium robertsii* DNA copies (c) in 87 of the 117 (74.4%) tested root samples. In detail, 41 of 44 (82.2%) roots inoculated with EF3.5(2) or ARSEF-2575-GFP, respectively, were tested positive. However, we also detected *M. robertsii* DNA in the roots of the control plants with 6 out of 28 (21.4%) plants being positively amplified by ddPCR.

A median of exp(4.30) c/μL (range 0–exp(6.04)) at 14 dpi, exp(4.20) c/μL (range 0–exp(7.60)) at 35 dpi, exp(2.74) c/μL (range exp(1.32)–exp(5.08)) at 63 dpi and exp(4.61) c/μL (range 0–exp(7.20)) at 96–98 dpi, respectively, was quantified in DNA extracted of root samples inoculated with the strain EF3.5(2). Meanwhile, *M. robertsii* DNA copies from roots of the ARSEF-2575-GFP treatment reached median concentrations of exp(3.55) c/μL (range 0–exp(5.20)) at 14 dpi, exp(3.60) c/μL (range exp(0.74)–exp(5.11)) at 35 dpi, exp(2.84) c/μL (range 0–exp(6.30)) at 63 dpi and exp(2.25) c/μL (range 0–exp(4.22)) at 96–98 dpi. Of the tested control plants, 6 positive amplifications were obtained, yet with very low copy numbers ranging from exp(0.3) to exp(2.9) c/μL. While significant differences between the two EIPF treatments occurred only at the end of the experiment at 96–98 dpi (pairwise Wilcoxon test, *p* = 0.0136), both treatments had higher EIPF concentrations than the control at all times (Kruskal–Wallis test, X^2^ = 13.496, df = 2, *p* = 0.001173; X^2^ = 14.044, df = 2, *p* = 0.0008922; X^2^ = 9.6779, df = 2, *p* = 0.007915 for 14, 35, 63 dpi, respectively) (Figure 1a). 

#### 3.1.2. Quantification of the Root Endophytic Potential of Two *M. robertsii* Strains

Considering the endophytic potential of *M. robertsii*, 51 of 115 tested root samples (44.4%), amplified positively in the ddPCR. *Metarhizium robertsii* DNA copies (c) were detected in 21 of 44 (47.8%) root samples treated with *M. robertsii* strain EF3.5(2), while the transformed strain ARSEF-2575-GFP was detected in 26 of 43 (60.5%) root samples. Among the control root samples, 4 out of 28 (14.3%) tested positive in the ddPCR.

The endophytic concentration of DNA from the root samples treated with either strain EF3.5(2) (median of exp(1.72) c/μL (range 0–exp(4.20))) or strain ARSEF-2575-GFP (median of exp(1.41) c/μL (range 0–exp(3.27))) was significantly higher compared to the controls at 35 dpi (Kruskal–Wallis test, X^2^ = 8.6207, df = 2, *p* = 0.01343), respectively). Similarly, endophytic root concentrations of *M. robertsii* differed between EIPF treatments and the control at 96–98 dpi (Kruskal–Wallis test, X^2^ = 10.784, df = 2, *p* = 0.004554), with a median exp(1.17) c/μL (range 0–exp(5.84)) for samples treated with the strain EF3.5(2) and exp(1.36) c/μL (range 0–exp(3.88)) for samples treated with the strain ARSEF-2575-GFP. However, no significant differences were detected at 14 dpi (Kruskal–Wallis test, X^2^ = 2.4277, df = 2, *p* = 0.2971) and 63 dpi (Kruskal–Wallis test, X^2^ = 0.15558, df = 2, *p* = 0.9252) (Figure 1b).

#### 3.1.3. Quantification of Endophytic Potential of Two *M. robertsii* Strains in *V. vinifera* Leaves and Stem

Regarding the potential of systemic colonization of grapevine by *M. robertsii* strain EF3.5(2), DNA was positively quantified in 12.2% (5/41) of the third leaves sampled, 9.3% (4/43) of the stem pieces, and 7.5% (3/40) of the terminal leaves, albeit with an overall low concentration range from exp(0.28) to exp(1.01) c/μL. In control plants, 3.6% (1/28) of the third leaves, 3.2% (1/31) of the stem pieces, and 9.6% (3/31) of the terminal leaves had a positive signal for *M. robertsii* DNA with a range of [0.28–1.5] exp(c/μL). In grapevines inoculated with the transformed *M. robertsii* strain ARSEF-2575-GFP, 26.3% (10/38) of the third leaves, 2.6% (1/39) of the stem pieces, and 28.2% (11/39) of the terminal leaves showed positive signals. Positive third and terminal leaf samples ranged from exp(0.36) to exp(3.35) c/μL with 5 plants scoring higher than exp(2.36) c/μL which is judged as a high value indicating a potential systemic colonization of the transformed strain ARSEF-2575-GFP. Results are summarized in Appendix A—Table A1.

### 3.2. Association of M. robertsii with V. vinifera Assessed via Culture-Based Method

Using the culture-based method *M. robertsii* strain EF3.5(2) and the transformed strain ARSEF-2575-GFP were classified as being associated with the rhizosphere in 65.9% (29/44) and 11.4% (5/44) of the grapevine plants, respectively, when all four post-inoculation periods were considered (Figure 2a). An association to the root endosphere was detected in 13.6% (6/44) of the tested grapevines for strain EF3.5(2) and 4.5% (2/44) for the transformed strain ARSEF-2575-GFP (Figure 2b). The strain EF3.5(2) was significantly associated with the rhizosphere of more grapevine plants than the transformed strain ARSEF-2575-GFP (Chisq.test: X^2^ = 184.94, df = 7, *p* < 2.2 × 10^−16^) relative to overall post-inoculation periods. However, there was no significant difference in the percentage of grapevines being endophytically associated with both strains (Chisq.test: X^2^ = 23.882, df = 7, *p* = 0.001195) according to the culture-based method. Additionally, the culture-based method did not reveal an evidence of systemic colonization of the plants as none of the tested leaves grown on the culture medium showed evidence of *M. robertsii* colonies (data not shown here).

The association of *M. robertsii* to the grapevine rhizosphere was significantly more often detected by ddPCR compared to the culture-based method for the transformed strain ARSEF-2575-GFP (Chisq.test: X^2^ = 30.556, df = 7, *p* = 7.507 × 10^−5^) but not for the strain EF3.5(2) (Chisq.test: X^2^ = 5.6571, df = 7, *p* = 0.5803). Same was true for the association of *M. robertsii* to the root endosphere which was higher when assessed by ddPCR compared to the culture-based method (strain EF3.5(2): Chisq.test: X^2^ = 15.429, df = 7, *p* = 0.03088; strain ARSEF-2575-GFP: Chisq.test: X^2^ = 32.571, df = 7, *p* = 3.181 × 10^−5^).

### 3.3. Observation of the M. robertsii Transformed Strain ARSEF-2575-GFP Association with Grapevine Roots

The microscopic results of grapevine roots inoculated with the transformed *M. robertsii* ARSEF-2575-GFP strain showed the successful adhesion of the fungus to the root surface as indicated by positive fluorescent signals at 14 dpi (Figure 3a). At 31 dpi spore germination was observed at the root surface (Figure 3b–d) with a few hyphae emerging from the spores attached to the root surface, which indicated the rhizospheric competence of this strain. 

## 4. Discussion

*Metarhizium* spp. strains were previously demonstrated to be rhizosphere and root endosphere colonizers of diverse plant species from annual to wild flowers, grasses, annual to perennial crops, shrubs and trees [6,9,16,24,29,59,60,61,62] with subsequent stimulation of the root growth in some cases [7,63]. The potential for association of *M. robertsii* with roots has been supported by studies of *Metarhizium* spp. demonstrating its natural association with the rhizosphere and root-endosphere of several plant species from diverse botanical families [6,30,59,64,65,66]. It was reinforced by culturing above and below ground tissue of 82 flower species randomly sampled in Canadian grassland, with 47 (57%) of the samples being colonized at the root level by *Metarhizium* spp. [6]. Based on results obtained via culture-based method we found that 85% of the inoculated grapevines were colonized at the rhizosphere level at 14 dpi by the strain EF3.5(2), dropping to 40% at 96–98 dpi compared to the respective 0% and 13% at both assessment dates for the strain ARSEF-2575-GFP. For the root endosphere, the colonization level ranked from 23% at 14 dpi to 10% at 96–98 dpi for the strain EF3.5(2) and from 0 to 13% for the strain ARSEF-2575-GFP. Previously, *M. robertsii* was found to endophytically establish in the roots of tomatoes with colonization levels of 100% to 95% from 10 to 30 dpi after seed inoculation [67]; in maize roots the level of endophyte colonization was 82% after seed-dressing [60]. A high level of endophytic colonization of cassava *Manihot esculenta* Crantz with three *M. anisopliae* strains was observed from 100% to 50% after 7–9 dpi and 100% to 20% after 47–49 dpi with root drenching inoculation [16]. The endophytic root colonization of two strawberry *Fragaria* × *ananassa* Duch varieties ranked from 20 to 100% at 180 dpi after plantlet dipping also depending on the tested strain [68]. Additionally, *M. anisopliae* succeeded high percentage of root colonization (83%) of *Vicia faba* L. 30 dpi after its seed-inoculation [32]. In comparison, the colonization of French bean roots was judged low by the authors with 30% of endophytic root colonization by *M. robertsii* at 35 dpi after seed immersion [69] compared to their previous results. [70] recorded no endophytic association of *M. anisopliae* with both common bean and French bean, 14 days post seed-dressing, and [71] found the same results using the same *M. anisopliae* strain inoculated to maize using seed-treatment. Thus, the endophytic root-colonization of grapevines in our trial by the two *M. robertsii* was minimal, which could be attributed to the use of a sterile soil substrate, as [72] found that *B. bassiana* endophytically colonized less leaves, stems and roots of sorghum *Sorghum bicolor* L. planted in sterile and non-sterile soil compared to vermiculite. Additionally, Ref. [19] found low endophytic root-colonization levels of *M. anisopliae* in the common bean planted in sterile and non-sterile soil, and concluded that sterility of the substrate had impaired the colonization capacity.

A previous study showed successful colonization of the EIPF *Beauveria bassiana* strain H2S32 with grapevine *V. vinifera* var. Sideritis both grafted on R110 rootstock and self-rooted after drenching with this fungus. The respective percentage of endophytic root colonization after 53 dpi ranked from 82.5% for the self-rooted plants to 80% for the grafted ones and the colonization had a subsequential enhancement of grapevine growth [46]. Additionally, it was recorded that 50% of tested grafted grapevines *V. vinifera* cv. Pinotage were endophytically colonized by *B. bassiana* at leaf level 21 days after root drenching [49]. Thus, EIPF have the potential of endophytic association with both grafted and un-grafted grapevines which can in turn evolve into systemic colonization.

Remarkably, evidence of systemic colonization of grapevine plants by *M. robertsii* strain ARSEF-2575-GFP was demonstrated via ddPCR but not via the culture-based method, with 6.5% (5/77) of tested leaves classified as endophytically colonized. Yet, a lot of studies focusing on *Metarhizium* spp. endophytic colonization did not record proof of colonization in the above-ground part of the trialed plant species, but only in the roots [6,7,16,29,32,69]. In one study, French bean *Phaseolus vulgaris* was drenched with a fungal suspension of the strain ARSEF-2575-GFP, resulting in 10% of stems and leaves being endophytically colonized with EIPF at 60 dpi [24]. However, a systemic colonization of the EIPF *Metarhizium* spp. is commonly observed when employing other methods of inoculation then root-drenching or direct contact with the soil [73]. The seed-treatment method usually induced high rates of systemic colonization by *Metarhizium* spp. during the initial phase of the trial. For example, 28.9% of maize leaf sections were observed to be endophytically colonized with *M. robertsii* strain after seed inoculation [60]. When tomato seeds were inoculated with *M. robertsii,* 60% of the sampled leaves were endophytically colonized at 10 dpi, decreasing to 20% at 30 dpi and from 20% to 5% in stems [67]. The foliar application of conidia of one leaf of rapeseed *Brassica napus* L. with *M. anisopliae* induced systemic colonization of the plants ranking from 50% to 80% for sampled leaves, 35% to 75% for the petioles, and finally 15% to 35% for the stems after 14 to 35 dpi [74]. Additionally, tomato, melon *Cucumis melo* L. var. *reticulatus* Naud and Alfalfa *Medicago sativa* L. were endophytically colonized by *Metarhizium* spp. with a rank of colonization from 65% to 35% in the leaves and 70 to 35% in the stems after 24 to 96 h post treatment [75].

In this study, the ddPCR revealed that both strains colonized the grapevine root rhizosphere to the same extent at 63 dpi. However, the culture-based method showed a higher percentage of rhizosphere colonization by the strain EF3.5(2) native to vineyard compared to the transformed strain ARSEF-2575-GFP during the time course of the experiment. This result was corroborated by quantification using ddPCR at the end of the experiment, which showed a higher persistence of association of the native strain. The reasons for this higher establishment in the grapevine rhizosphere could be related to the propensity of the strain EF3.5(2) to exploit the photosynthetic products secreted by grapevine in the soil [10]. Indeed, *M. robertsii* rhizosphere-competence genes are up-regulated when in contact with high concentration of root exudates secreted by the surrounding plant. Additionally, the success of one fungal strain to colonize the rhizosphere is directly related to its capacity to utilize secreted plants metabolites, such as sucrose [76]. It was demonstrated that some species from the *M. anisopliae* complex, including *M. robertsii* strain ARSEF-2575-GFP have a better ability to grow and germinate at high concentrations of root exudates [10]. As the strain ARSEF-2575-GFP is non-native to the vineyard soil, it might lack specialization of specific grapevine excreted root exudates compared to the strain EF3.5(2), thus impairing its ability to multiply in the rhizosphere and colonize the roots. Thus, the vineyard native strain seems to be the best candidate as a biocontrol agent to protect grapevine roots against soil pests because it shows a high persistence of association with grapevine roots.

Both strains were found to endophytically establish inside grapevine roots with the same pattern according to ddPCR and culture-based methods, yet with quite a low incidence of colonization. The endophytic *M. robertsii* DNA quantification in grapevine roots inoculated either with *M. robertsii* EF3.5(2) or the strain ARSEF-2575-GFP did not differ significantly from the quantification in the control plants at 14 and 63 dpi. This could be explained by the fact that the establishment of *M. robertsii* inside roots happened independently of the experimental timing. As a result, more plants from the pool of replicates screened at 35 dpi were endophytically colonized compared to the ones at 63 dpi. Similar results were obtained when quantifying endophytic French bean root colonization by *M. robertsii* strain ARSEF-2575-GFP [24]. The amount of *M. robertsii* DNA quantified decreased from 3 to 10 dpi and then increased from 10 to 14 dpi. These quantifications variations could be explained by the number of plants colonized at the respective time of detection. Additionally, when quantifying an artificially inoculated endophyte, *Serendipita herbamans* associated with the knotweed *Reynoutria* ssp. roots, high variations of quantification were found particularly under suitable conditions of establishment [77].

Comparing the different methods used in this study, ddPCR is a sensitive method which can precisely detect and quantify microorganisms from various environments even if they are present at very low concentrations [78], with more precision and less technical preparation than qPCR or nested qPCR [26,27,28,79,80]. It has the best quantification performance compared to other DNA quantification methods for samples taken from environments with complex matrix prone to PCR inhibition and rich of non-target DNAs like soil or plant tissue [81]. These attributes of ddPCR make it an insightful tool for DNA quantification of endophytes in plants allowing better characterization of endophyte’ association with plants and the kinetics of establishment. However, the high sensitivity of the method may consequently influence its specificity by detecting very low levels of DNA. In our study, the presence of *M. robertsii* was detected in roots, leaves and stems of control plants which might originate from other microorganisms with similar DNA sequences [82]. Thus, ddPCR is also prone to detect false positives, depending on primer specificity.

To characterize the endophytic potential of *M. robertsii* in grapevine plants, we also employed a more traditional culture-based method in our study. This method is most commonly used as it is affordable, easy to handle with standard microbiology instruments, and is rapid with on average 10 days necessary to observe the fungal growth on the plates [22]. In contrast, ddPCR is a costly molecular tool which requires sophisticated PCR equipment, some pre-testing for preparation of samples and significantly extra work with DNA extraction. The culture-based method has the main advantage to characterize the microorganism viability as only the live propagules grow on the medium, as opposed to ddPCR, which quantifies DNA from viable and non-viable cells complicating scientific interpretations on the quantified inoculum [83]. In addition, the culture-based method is not suitable for fungal propagules quantification because the plating does not assure homogeneous distribution of fungal spores or the respective inoculum has a low vitality [27]. Additionally, the competitiveness of other endophytic species that grow abundantly on the respective growth medium because of its lack of selectivity can bias the quantification of individuals which are over or under represented [23]. The ddPCR is more sensitive than the culture-based method used to detect fungal endophytes, which was apparent in our study when detecting both *M. robertsii* strains in the root endosphere and in the aboveground parts of grapevine. ddPCR characterized several leaf and stem tissue samples as being endophytically colonized with both *M. robertsii* strains, while culture-based method showed no evidence of systemic colonization of grapevine by the respective strains.

Confocal microscopy was used to complete the range of methods employed in this study. Compared to the two other methods discussed so far, it gives the most irrefutable proof of endophytic and rhizosphere colonization of the plant [24]. It is the only method that characterizes the fungal distribution inside the plant tissues as well as the fungal structures colonizing the plant [6,7]. Additionally, a transformed strain with green fluorescent protein is usually inoculated for microscopic tracking, making it an undisputable proof of the colonization potential of the fungus [53]. However, a significant amount of time is necessary to create the transformant strain, to screen a large number of plant replicates and tissues of a single plant, and to prepare a significant number of microsections of different tissues. Additionally, the chances of detecting endophytic colonization with confocal microscopy are low compared to the ddPCR or the culture-based method, making the use of these more sensible methods mandatory for endophytic detection surveys [84]. The results of our study are in line with this finding, we found no evidence of endophytic colonization of the transformed strain ARSEF-2575-GFP by confocal microscopy observation while ddPCR and culture-based method detected endophytic colonization. However, further confocal microscopy observations with more screened root segments should be made to confirm the obtained results.

## 5. Conclusions

The present study is the first to demonstrate the power of a combination of methods used to investigate endophytic establishment of *M. robertsii* in grapevine. We found a significantly greater establishment of the vineyard native *M. robertsii* EF3.5(2) strain compared to the GFP-transformed strain ARSEF-2575-GFP in the rhizosphere of grapevine. This could potentially be explained by the specific affinity of the native strain to the root exudates of grapevine enhancing its multiplication in the rhizosphere.

ddPCR is the most sensitive detection method and we recommend to include it in studies aiming to characterize the kinetics of endophytic fungal establishment and their systemic colonization. Nonetheless, the method is not a stand-alone technique, but should be accompanied by culture-based and/or confocal microscopy that provides addition information on the viability of the fungus.

## Figures and Tables

**Figure 1 microorganisms-10-02437-f001:**
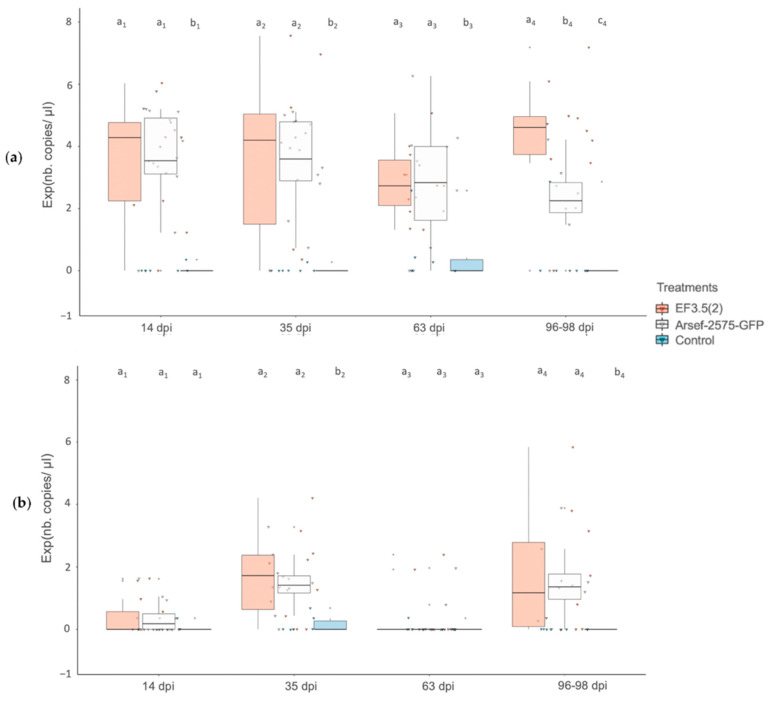
(**a**) Time-course of quantification of rhizospheric *V. vinifera* association of two *M. robertsii* strains (EF3.5(2), red boxes; GFP transformed strain ARSEF-2575-GFP, white boxes) and a control treatment (blue boxes) using ddPCR. The boxplots represent the logarithm of the number (nb) of DNA copies (of non-null values) of *M. robertsii* per microliter of DNA extracted of mixed non-disinfected grapevine roots with adhering soil (rhizosphere) 14-, 35-, 63- and 96–98 days post inoculation (dpi). Significant differences are indicated by small letters above to boxes (Kruskal–Wallis test, *p* < 0.05); (**b**) Time-course of quantification of endophytic *V. vinifera* root association of two *M. robertsii* strains using ddPCR. The boxplots represent the logarithm of the number (nb) of DNA copies (of non-null values) of *M. robertsii* per microliter of DNA extracted of mixed disinfected grapevine roots (root endosphere) 14-, 35-, 63- and 96–98 days post inoculation (dpi).

**Figure 2 microorganisms-10-02437-f002:**
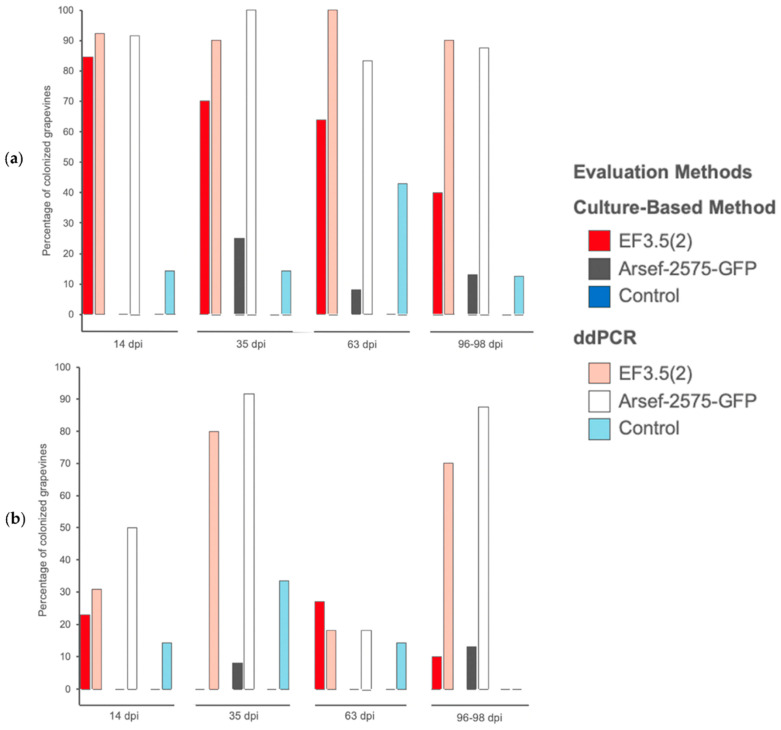
Time-course of detection of two *M. robertsii* strains (EF3.5(2); GFP-transformed strain ARSEF-2575-GFP) and a control treatment on *V. vinifera* roots. The bars represent the percentage of *M. robertsii* colonized grapevines at (**a**) the rhizosphere and (**b**) the root endosphere evaluated with two methods: culture-based method (EF3.5(2): red bars, ARSEF-2575-GFP: grey bars, controls: dark blue bars) and ddPCR (EF3.5(2): pink bars, ARSEF-2575-GFP: white bars, controls: light blue bars). Evaluation was made 14-, 35-, 63- and 96–98 days post inoculation (dpi).

**Figure 3 microorganisms-10-02437-f003:**
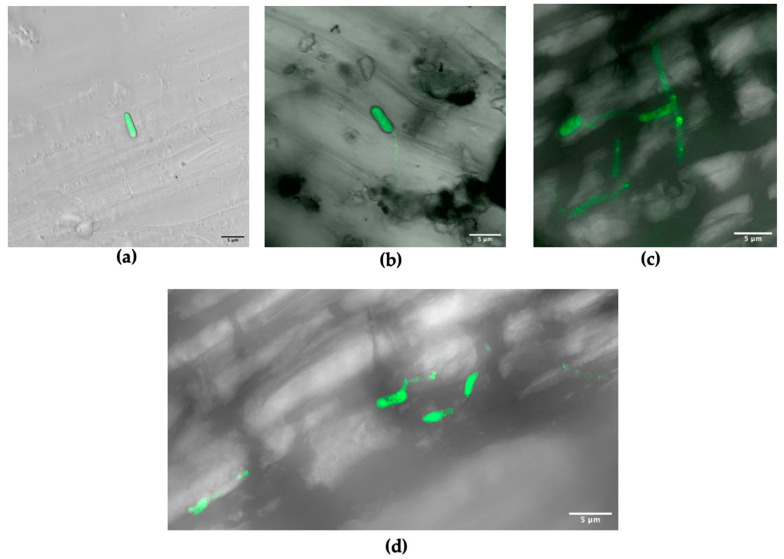
Confocal images of grapevine root association with *M. robertsii* ARSEF-2575-GFP expressing green fluorescent protein (GFP) observed at (**a**) 14 dpi, and at (**b**–**d**) 31 dpi. All scale bars are 5 μm and total magnification was ×63 using an oil-immersion objective.

## Data Availability

Not applicable.

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
