# Peer review of "Three Methods Assessing the Association of the Endophytic Entomopathogenic Fungus Metarhizium robertsii with Non-Grafted Grapevine Vitis vinifera"

_microorganisms, 2022, doi:10.3390/microorganisms10122437_

Round 1

Reviewer 1 Report

Comments and Suggestions for Authors

Manuscript ID: microorganisms-2049998

The paper entitled “Association of the endophytic entomopathogenic fungus Metarhizium robertsii with non-grafted grapevine Vitis vinifera” was carefully reviewed. The combination of quantitative and qualitative data achieved by both molecular and microscopy methods are largely used to monitor endophytes, with the advantages of each technique complementing the drawbacks of the other. In the present manuscript, the authors describe a combination of molecular and microscopy techniques to ascertain the mechanisms of non-grafted grapevine colonization by M.  robertsii isolates (i.e., ddPCR, culture-based method, and confocal imaging). 

The manuscript still needs major revisions before considering publication in “Microorganisms”.

Limitations of the work, novelty, and contributions should be highlighted more: This manuscript is an interesting laboratory study to demonstrate the rhizospheric and endophytic potential of M. robertsii with non-grafted grapevine V. vinifera after artificial inoculation. However, the paper is not at all novel outside of where it is focused upon. It is always best to carry out efficiency tests in the field (effects of environmental conditions on endophyte colonization). For example, a meta-analysis of published studies (years 2002–2018) on the percentage of Beauveria bassiana plant colonization across the plant kingdom was conducted. studies conducted in controlled environments resulted in higher endophytic colonization with B. bassiana than field studies. Then, I strongly recommend the authors to conduct field experiments to confirm and validate the previously obtained results in the laboratory.  

Detailed comments:

Keywords

-          In general, avoid using keywords that are already in the title. Replace “Association, Grapevine, …”.

Abstract

-          Line 16: Indicate the full scientific name of the fungus.

-          Add quantitative results to this section.

-          Improve on the conclusions of the abstract.

Introduction

-          This section still needs to be improved. Enlarge the state of the art by adding other relevant and recent works in the field.

-          Please write one or two paragraphs to present the endophytic mycota associated with Vitis vinifera.

-          The novelty of the work is not clear from the Introduction. Rewrite the last paragraph.

-          Lines 93-96: The Latin binomial should be indicated as follows: the name of the genus, the specific epithet, and the authority.

Materials and methods

-          Lines 151-152: The authors used a concentration of 1 x 107 conidia/mL to inoculate potted grapevine plants with M. robertsii strains. Can you explain why you specifically chose this concentration? Please provide a reference for this.

-          Lines 223-226 (Statistical analysis): The authors indicated that “For the ddPCR quantification from samples of the rhizosphere and the root endosphere data were analyzed separately at each tested time point post inoculation to compare the two tested strains to the control treatment”.

I have major reservations about the data analysis. Which tests were used to determine the differences among the strains? 

Results:

-          Figure 3: I've noticed that the authors directly labelled Figure 3 as 3b, 3c, and 3d. Could you label the sections sequentially as 3a, 3b, 3c and 3d. Please improve the resolution of the figure as the four images are not clear.

Discussion:

-          The quality of the discussion of the results has to be improved. I suggest adding other relevant and recent works in the field as, for instance:

1.         González, V.; Tello, M.L. The Endophytic Mycota Associated with Vitis Vinifera in Central Spain. Fungal Divers. 2011, 47, 29–42, doi:10.1007/s13225-010-0073-x.

2.         Hu, S.; Bidochka, M.J. Root Colonization by Endophytic Insect‐pathogenic Fungi. J. Appl. Microbiol. 2021, 130, 570–581, doi:10.1111/jam.14503.

3.         Resquín-Romero, G.; Garrido-Jurado, I.; Delso, C.; Ríos-Moreno, A.; Quesada-Moraga, E. Transient Endophytic Colonizations of Plants Improve the Outcome of Foliar Applications of Mycoinsecticides against Chewing Insects. J. Invertebr. Pathol. 2016, 136, 23–31, doi:10.1016/j.jip.2016.03.003.

4.         Batta, Y.A. Efficacy of Endophytic and Applied Metarhizium Anisopliae (Metch.) Sorokin (Ascomycota: Hypocreales) against Larvae of Plutella Xylostella L. (Yponomeutidae: Lepidoptera) Infesting Brassica Napus Plants. Crop Prot. 2013, 44, 128–134, doi:10.1016/j.cropro.2012.11.001.

5.         González-Pérez, E.; Ortega-Amaro, M.A.; Bautista, E.; Delgado-Sánchez, P.; Jiménez-Bremont, J.F. The Entomopathogenic Fungus Metarhizium Anisopliae Enhances Arabidopsis, Tomato, and Maize Plant Growth. Plant Physiol. Biochem. 2022, 176, 34–43, doi:10.1016/j.plaphy.2022.02.008.

6.         Krell, V.; Jakobs-Schoenwandt, D.; Vidal, S.; Patel, A.V. Encapsulation of Metarhizium Brunneum Enhances Endophytism in Tomato Plants. Biol. Control 2018, 116, 62–73, doi:10.1016/j.biocontrol.2017.05.004.

7.         Krell, V.; Unger, S.; Jakobs-Schoenwandt, D.; Patel, A.V. Importance of Phosphorus Supply through Endophytic Metarhizium Brunneum for Root:Shoot Allocation and Root Architecture in Potato Plants. Plant Soil 2018, 430, 87–97, doi:10.1007/s11104-018-3718-2.

8.         Wagner, B.L.; Lewis, L.C. Colonization of Corn, Zea Mays, by the Entomopathogenic Fungus Beauveria Bassiana. Appl. Environ. Microbiol. 2000, 66, 3468–3473.

9.         Qian, X.; Li, H.; Wang, Y.; Wu, B.; Wu, M.; Chen, L.; Li, X.; Zhang, Y.; Wang, X.; Shi, M.; et al. Leaf and Root Endospheres Harbor Lower Fungal Diversity and Less Complex Fungal Co-Occurrence Patterns Than Rhizosphere. Front. Microbiol. 2019, 10.

10.       Garnica, S.; Liao, Z.; Hamard, S.; Waller, F.; Parepa, M.; Bossdorf, O. Environmental Stress Determines the Colonization and Impact of an Endophytic Fungus on Invasive Knotweed. Biol. Invasions 2022, 24, 1785–1795, doi:10.1007/s10530-022-02749-y.

11.       Kiarie, S.; Nyasani, J.O.; Gohole, L.S.; Maniania, N.K.; Subramanian, S. Impact of Fungal Endophyte Colonization of Maize (Zea Mays L.) on Induced Resistance to Thrips- and Aphid-Transmitted Viruses. Plants 2020, 9, doi:10.3390/plants9040416.

12.       Andreolli, M.; Zapparoli, G.; Lampis, S.; Santi, C.; Angelini, E.; Bertazzon, N. In Vivo Endophytic, Rhizospheric and Epiphytic Colonization of Vitis Vinifera by the Plant-Growth Promoting and Antifungal Strain Pseudomonas Protegens MP12.Microorganisms 2021, 9, 234, doi:10.3390/microorganisms9020234.

-          In addition, the authors did not compare their results in depth with others reported in the literature. I recommend that the authors rewrite the discussion section.

-          Lines 419-423: This information has already been included in the results section; delete this paragraph to avoid redundancy.

Finally, the paper would really benefit from the proof reading by an English speaker as it contains numerous typos.

Overall, this manuscript is fairly well written; after addressing the above comments, it may be considered for publication.

Reviewer 2 Report

A nice paper that can be published (nearly) as is.

Minor comments:

Line 299: "M. robertsii" should be in italics

Line 445: Values like 81.64% should be rounded to i.e. 82% due to significance

Round 2

Reviewer 1 Report

The edits / revisions from the previous manuscript were well implemented, and the paper flows very nicely now.